

# A change in the relationship between ENSO and the South Atlantic Subtropical Dipole in the past four decades

Lejiang Yu[1*], Shiyuan Zhong[2], Timo Vihma[3], Cuijuan Sui[4], and Bo Sun[1]

MNR Key Laboratory for Polar Science, Polar Research Institute of China,

Shanghai, China,

Department of Geography, Environment and Spatial Sciences, Michigan State

University, East Lansing, MI, USA,

Finnish Meteorological Institute, Helsinki, Finland
National Marine Environmental Forecasting Center, Beijing, China

*Corresponding Author's address Dr. Lejiang Yu

MNR Key Laboratory for Polar Science, Polar Research Institute of China, Shanghai,

China

Jinqiao Road 451, 200136, Shanghai, China

Phone: 0086-020-58712034, email: yulejiang@sina.com.cn



**Abstract**
This study investigates the relationship between sea surface temperature (SST) in the
subtropical Atlantic Ocean, as represented by the Southern Atlantic Subtropical
Dipole (SASD), and SST in the tropical Pacific Ocean, identified by the El
Niño-Southern Oscillation (ENSO). Our analysis reveals a significant inverse
correlation between the SASD and Niño indices over a century, with multi-decadal
variability that contradicts weak simultaneous correlations previously reported in the
literature. The study also highlights a strengthening of their inverse correlations in the
most recent two decades compared to the preceding two decades, which can be
attributed to the shift in ENSO regime from more frequent eastern Pacific El Niño to
central Pacific El Niño around the turn of the century. This shift helps set the stage for
changes in convective activity in the critical region (20°S-40°S, 180°-140°W) of the
central South Pacific Ocean, triggering wavetrains that propagate along different paths
and ultimately contributing to different southern Atlantic subtropical high (SASH) and
changes in anomalous SST patterns in the subtropical Atlantic Ocean. These findings
advance our understanding of the interactions between South Atlantic and Pacific SST
variations, which strongly influence rainfall patterns particularly in South America
and southern Africa and may improve sub-seasonal to seasonal precipitation
predictions in these regions.
**Plain Language Summary**
We look at how changes in sea surface temperature (SST) in the subtropical Atlantic



Ocean and the tropical Pacific Ocean are related to each other over time. We found
that there is a significant inverse correlation between the two areas over the course of
a century, and that this relationship changes over time in multi-decade cycles. We also
found that the inverse relationship has become stronger in the recent two decades
compared to the proceeding two decades. This change is likely due to a regime shift in
ENSO around the turn of the century that helps set stage to the occurrences of
different convective activities in a particular area of the central South Pacific Ocean,
ultimately leading to changes in SST in the subtropical South Atlantic through
atmospheric waves. Understanding these relationships is important because they can
affect, among other things, rainfall in South America and southern Africa, and the
study may help improve predictions of future rainfall patterns in these regions.

**Key Points**
In contrary to the current understanding, there can be a strong connection between
ENSO and the South Atlantic Subtropical Dipole (SASD).
It is highly probable that the robust inverse correlation between ENSO and SASD will
persist in the future.
The ENSO-SASD correlation exhibits substantial multi-decadal variability over the
course of a century.
The change in the ENSO-SASD relation can be linked to changes in ENSO regime
and convective activities over the central South Pacific Ocean.





## 1. Introduction

The South Atlantic subtropical dipole (SASD) mode represents an opposite variability of sea surface temperature (SST) over the northeastern and southwestern South Atlantic Ocean (Venegas et al., 1997; Sterl and Hazeleger, 2003; Haarsma et al., 2005; Morioka et al., 2011). The SASD exhibits a strong seasonal variability that peaks in austral summer and is related to the strength and location of the South Atlantic subtropical high (SASH). The anomalous surface circulations associated with SASH influence the oceanic mixed layer depth and surface evaporation, leading to SST anomalies over South Atlantic subtropical regions (Sterl and Hazeleger, 2003; Haarsma et al., 2005; Colberg and Reason, 2006; Morioka et al., 2011). Surface shortwave radiation also plays a vital role in the formation and decay of the SASD (Morioka et al., 2011).

The SASD also shows an interannual variability, which has been linked to year-to-year changes in precipitation in southern Africa (Vigaud et al., 2009; Morioka et al., 2011), West Africa (Nnamchi and Li, 2011) and South America (Muza et al., 2009; Bombaridi and Carvalho, 2011; Kayano et al., 2013; Bombardi et al., 2014). Understanding what factors are behind the SASD interannual variability may improve precipitation predictions for these regions. Previous studies have investigated the effects of large-scale climate indices on the SASD. Hermes and Reason (2005) and Morioka et al. (2014) noted the effect of the Antarctic Oscillation (AAO) on the SASD. Several studies have confirmed the linkage between the SASD and the Subtropical Indian Ocean Dipole (SIOD) (Fauchereau et al., 2003; Hermes and





Reason, 2005; Lin, 2019; Yu et al., 2023). A weak relationship has been noted
between the simultaneous values of SASD and ENSO at zero lag (Venegas et al., 1997;
Hermes and Reason, 2003; Fauchereau et al., 2003; Kayano et al., 2013). However,
Kayano et al. (2013) detected a significant lagged correlation between the two indices.
Rodrigues et al. (2015) explained the weak SASD-ENSO relationship by showing that
although negative (positive) SASD events are associated with the positive (negative)
phase of the central Pacific El Niño events, such association is absent during eastern
Pacific El Niño events.
The weak simultaneous and strong lagged relationships between SASD and ENSO
have been established based on data for earlier time periods prior to 2010 (Rodrigues
et al., 2015). However, since 2010, there have been a major El Niño event and two
record-breaking La Niña events, making the last decade invaluable for determining
how robust the established SASD-ENSO relationship is. In this study, we have
extended previous analyses to include the most recent decade with the aim to test the
robustness of the SASD-ENSO relationship, and to offer additional insight into the
mechanisms behind the teleconnection between the SST anomalies in the tropical
Pacific and subtropical South Atlantic Oceans. In addition, we have expanded our
analysis by examining SST time series over a course of a century dating back to 1871.

**2. Datasets and methods**
The monthly SST data applied in this study are from the U.S. National Oceanic
and Atmospheric Administration (NOAA) Extended Reconstructed Sea Surface





Temperature (ERSST) version 5 (Huang et al., 2017) covering the globe with a
horizontal resolution of 2.0 ºlatitude × 2.0 ºlongitude and spanning from 1871 to the
present. The primary focus of our study is the recent four decades from 1979 through
2020. Following Morioka et al. (2011), we define the SASD index as the difference of
the SST anomalies between the south-western (10-30$^o$ W, 30-40$^o$ S) and north-eastern
(0-20$^o$ W, 15-25$^o$ S) South Atlantic Ocean (Figure 1a). The ENSO signal is represented
primarily by the Niño 3.4 index, defined on the basis of SST anomalies in the tropical
central Pacific Ocean (5$^o$ N-5$^o$ S, 120-170$^o$ W) (Trenberth, 1997). The relationship
between ENSO and SASD is determined by the 18-year sliding correlation between
the detrended time series of the SASD index and the Niño 3.4 index. Two other ENSO
indices, the Niño 3 index (5$^o$ N-5$^o$ S, 90-150$^o$ W) and the Niño 4 index (5$^o$ N-5$^o$ S, 160
$^o$ E-150$^o$ W), are also included in our analyses to test the sensitivity of the
SASD-ENSO relationship to the types of ENSO events. The statistical significance of
the correlation is determined by the two-tailed student's t test and the statistical
significance of the difference between two time series of regression coefficients is
tested by the Z-test (Clogg et al., 1995).
We explore the mechanisms underlying the SASD-ENSO relationship through
examining the corresponding atmospheric circulations using monthly data from the
European Centre for Medium-Range Weather Forecasts (ECMWF) fifth-generation
reanalysis (ERA5) that has a 1/4 degree latitude and longitude horizontal resolution
and covers the period from 1979 to the present (Hersbach et al., 2020). We
supplement the ERA5 data with the monthly NOAA Interpolated Outgoing Longwave





Radiation (OLR) data, which has a horizontal resolution of 2.5 ° latitude × 2.5 °
longitude (Liebmann and Smith, 1996). To analyze atmospheric planetary waves, we
calculate the 200-hPa Rossby wave source (RWS) using the formulae proposed by
Sardeshmukh and Hoskins (1998):

$\text{RWS} = -V_\chi \cdot \nabla \zeta - \zeta D \approx -V_\chi' \cdot \nabla \bar{\zeta} - \bar{\zeta} D'$                                    (1)

wherein $\zeta$ is the vertical component of the absolute vorticity, $V_\chi$ and $D$ are the
divergent wind and divergence at  200 hPa, respectively. Climatological mean and
perturbation are represented by overbar and prime, respectively.

Additionally, we also analyze 200-hPa wave activity flux (WAF), which is derived

using the equation proposed by Takaya and Nakamura (2001):

$$W = \frac{p \cos \phi}{2|U|} \begin{pmatrix} \frac{\bar{u}}{a^2 \cos^2 \phi}\left[\left(\frac{\partial \psi'}{\partial \lambda}\right)^2 - \psi'\frac{\partial^2 \psi'}{\partial^2 \lambda}\right] + \frac{\bar{v}}{a^2 \cos \phi}\left[\frac{\partial \psi'}{\partial \lambda}\frac{\partial \psi'}{\partial \phi} - \psi'\frac{\partial^2 \psi'}{\partial \lambda \partial \phi}\right] \\ \frac{\bar{v}}{a^2 \cos \phi}\left[\frac{\partial \psi'}{\partial \lambda}\frac{\partial \psi'}{\partial \phi} - \psi'\frac{\partial^2 \psi'}{\partial \lambda \partial \phi}\right] + \frac{\bar{v}}{a^2}\left[\left(\frac{\partial \psi'}{\partial \phi}\right)^2 - \psi'\frac{\partial^2 \psi'}{\partial^2 \phi}\right] \end{pmatrix}$$

(2)

Where $\lambda$ and $\phi$ are longitude and latitude, respectively; $\psi$ is geostrophic stream
function; $|U|$ is climatological horizontal wind; $\bar{u}$ and $\bar{v}$ are the climatological zonal
and meridional winds, respectively; $p$ is the pressure divided by 1000 hPa; a is the
earth's radius. Overbar symbols represent climatological values while prime symbols
denote anomalies.

All the analyses presented focus on austral summer when both ENSO and SASD

peak in the annual cycle. Here, austral summer refers to January, February and March
(JFM) as SASD tends to peak in February (Morioka et al., 2012).



To further corroborate our statistical findings, we conduct numerical experiments
using Version 5 of the Community Atmosphere Model (CAM5), which serves as the
atmospheric component of the Community Earth System Model (CESM). CAM5
features 30 vertical levels and a horizontal resolution of 1.9 latitude $\times$ 2.5 longitude.
For more in-depth information about CAM5, please refer to Neale et al. (2011).
Several numerical experiments are conducted. These experiments include a
control run, which spans 50 years and is forced by the climatological annual cycle of
SST and sea ice concentration data from the Hadley Center, and three idealized
experiments where a +2 ℃ SST anomaly is introduced in the Southern Pacific region
(20°S-40°S, 180°-140°W), the Niño4 region over the central tropical Pacific (5°N-5°S,
160°E-150°W) and the Niño3 region over the eastern tropical Pacific (5°N-5°S,
150°W-90°W), respectively. The warm SST anomaly is introduced annually from
January 1 to March 31, while the SST and sea ice conditions in the remaining months
and regions are maintained at climatological levels. The last 20 years of the control
run are utilized as the reference years to restart the idealized experiments. The results
from the numerical experiments are presented as the differences between the idealized
experiments and the control experiment.


**3. Results**
3.1 Variability of the SASD-ENSO relationship
In contrast to the weak simultaneous correlation between the SASD and the Niño



3.4 indices suggested in previous studies, our results indicate a significant inverse
correlation (r = -0.51, p<0.01) between the two indices (Figure 1b, c) over the period
of 1979 through 2020. The magnitudes of the 18-year moving correlation coefficients
of the detrended SASD index (Figure 1b) with the detrended Niño 3.4 index (Figure
1b) tripled from r ≈ -0.25 in the late-1980s to r ≈ -0.75 during 2006-2011, intercepting
the p = 0.05 line between 1993 and 1994 (Figure 1c). When we divided the four
decades into two equally long periods, we found that the correlation coefficient was
-0.32 (p > 0.05) for the period before 2000 and -0.77 (p < 0.01) for the period after
2000. Despite replacing Niño 3.4 with Niño 4 or Niño 3 indices, the marked
difference in correlation strength between the earlier and latter periods remains
unchanged, which is unsurprising given the known correlation among the three Niño
indices. Specifically, the correlation coefficients for Niño 4 and Niño 3 increased from
-0.34 and -0.10 (p > 0.05), respectively, for the earlier period to -0.71 (p < 0.01) for
both indices for the latter period. The overall correlations for the entire 42-year period
were -0.4 for Niño 4 and -0.48 for Niño 3 (p < 0.05), which is slightly weaker than the
correlation of -0.51 for Niño 3.4. While Kao and Yu (2009) suggested that there are
superior alternatives to represent ENSO diversity than the Niño 3, 3.4 and 4 indices,
the above results reveal, for the first time to our knowledge, a strong multi-decadal
variability in the SASD-ENSO relationship. In other words, the interannual SST
variability modes in the tropical Pacific and subtropical Atlantic Oceans are
teleconnected, but the teleconnection is highly variable on the decadal to
multi-decadal scale.





Recognizing that four decades may not provide a comprehensive understanding of
decadal to multi-decadal variability, we expanded our analysis to more than a century,
spanning from 1871 to 2020. As depicted in Figure S1, our century-long time series
analysis reveals a noticeable multi-decadal variability in the 18-year moving
correlation, which further validates our findings.

3.2.  Potential forcing mechanisms underlying the variability in the SASD-ENSO

relationship in the past four decades

We next explore what might be behind the decadal variability of the SASD-ENSO
relationship (Figure 1). Since Niño 3 and Niño 4 have yielded results similar to those
of Niño 3.4, the analyses here utilize Niño 3.4 only. We compare regressions of
anomalous SST and atmospheric fields onto the Niño 3.4 index for the 2000-2020
period with those for the 1979-1999 period (Figures 2-4). For SST, the regression
pattern appears to resemble more strongly the negative phase of the SASD index
during 2000-2020 than during 1979-1999 (Figure 2a, b), seen as larger and more
significant SST anomalies in the positive center of SASD during 2000-2020.
Subtracting the 1979-1999 pattern from the 2000-2020 pattern yields a La Niña
structure, where the values are negative in the eastern tropical Pacific but positive in
the central and western tropical Pacific (Figure 2c), suggesting a shift to more central
Pacific El Niño during the 2000-2020 period from more eastern Pacific El Niño
during the 1979-1999 period.
Significant differences in the SST regression patterns indicating changes in ENSO





regime between the two periods are accompanied by substantial differences in the
regression patterns of the top-of-the-atmosphere outgoing longwave radiation (OLR)
and atmospheric circulations (see Figure 3). During the 1979-1999 period, there are
negative OLR anomalies over the tropical Pacific Ocean and southeastern Pacific
Ocean and positive anomalies over the southwestern Pacific Ocean and northern
South America (Figure 3a). The negative OLR anomalies, an indication of more and
stronger convective activities, produce positive Rossby Wave Source or RWS
(Sardeshmukh and Hoskins, 1998) and upper-tropospheric divergent wind over the
tropical and the southeastern Pacific Ocean and the opposite OLR anomalies generate
negative RWS and upper-tropospheric convergent wind over the southwestern Pacific
Ocean (Figure 3d). The sign of RWS determines the sign of vorticity and the direction
of rotation in atmospheric waves, including Rossby waves. Specifically, positive
(negative) RWS corresponds to positive (negative) relative vorticity or
counterclockwise (clockwise) rotation of the upper atmosphere relative to the Earth's
surface in the southern hemisphere. According to Sardeshmukh and Hoskins (1998),
the components of RWS that represent the changes in vorticity include the advection
of vorticity and vorticity anomalies related to the divergence of the flow. These
vorticity anomalies propagate through teleconnection wavetrains. The negative RWS
anomalies over the southwestern Pacific Ocean trigger a wavetrain that propagates
southeastwards into the Ross Sea, then eastwards into the Amundsen, Bellingshausen,
and Weddell Seas, as depicted by the anomalous fields of WAF and 200-hPa
geopotential heights in Figure 4a. The wavetrain generates positive anomalies of the

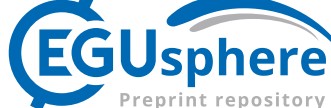

mean sea level pressure (MSLP) over the South Atlantic north of 40$^o$S and negative
anomalies over the southeastern South Atlantic (Figure 4b).
The negative anomalous MSLP and cyclonic circulation result in a negative wind
curl, which generates Ekman upwelling, leading to negative SST anomalies; the
opposite occurs for positive MSLP and anticyclonic circulation (Chaves and Nobre,
2004). The negative SST anomalies along Brazil coasts are also related to coastal
upwelling due to anomalous northeasterly winds (Franchito et al., 2008). Anomalous
warm (cold) air advection induced by anomalous northerly (southwesterly) winds
over the western South Atlantic Ocean also favors the formation of warm (cold) SST
centers of the negative phase SASD through downward (upward) transfer of sensible
heat flux (Figure 4b and 4d). Latent heat flux also plays an important role in the SST
anomalies (Barreiro et al., 2004). The enhanced westerly wind anomalies also lower
SST, which favors the development of the negative SST anomaly centers of the
negative phase SASD. Downward longwave radiation also shows a dipole structure,
which contributes to SST anomalies of the negative phase SASD (Figure S2a).
Compared to 1979-1999, the 2000-2020 period shows westward expansion of the
positive OLR anomalies into the tropical western Pacific (Figure 3b). There is an
increase in the extent, but a decrease in the magnitude, of the positive OLR anomalies
over the southwestern Pacific Ocean, and an increase in the significance of the
negative OLR anomalies over the central South Pacific Ocean. The OLR difference
between the two periods shows a La Niña state (Figure 3c). Negative OLR anomalies
indicating stronger than normal convective activities produce negative RWS and



upper-tropospheric convergence over the region east of New Zealand (Figure 3e),
which excites a wavetrain propagating eastwards into South Atlantic Ocean (Figure
4c). Under the influence of this wavetrain, the anomalous MSLP field displays a
dipole structure over the South Atlantic Ocean (Figure 4d). Similar to the discussion
about the earlier period, the horizontal heat advection associated with the anomalous
wind field, vertical heat transfer related to anomalous wind curl, and increased
westerly winds all play an important role in the formation of the negative phase SASD.
Weaker northeasterly wind anomalies relative to the earlier period weaken upwelling,
leading to warmer SST anomalies. The stronger easterly wind anomalies over the
eastern South Atlantic (25°S-30°S) offset the climatological southeasterly winds,
contributing to the warm SST anomalies. Downward longwave radiation anomalies
make larger contributions to the SST anomalies of the negative SASD than those over
the former period (Figure S2).
The distinct regression patterns of the OLR, RWS, and upper-level divergent
winds onto the Niño 3.4 index over the two periods (Figure 3c, f) suggest that the
varying strengths of convective activities over the central South Pacific Ocean may
have played an important role in the differences observed in the SASD-Niño 3.4
correlations between the two periods by triggering wavetrains propagating along
different paths. Over the key region (20-40°S, 140-180°W) in the central South
Pacific (green rectangle in Figure 3c), the correlation coefficients between the SASD
index and the OLR anomalies are -0.16 (p > 0.05) for the 1979-1999 period and -0.66
(p < 0.01) for the 2000-2020 period. A comparison of the climatological SST and





OLR in this key region between the two periods shows a significant increase in SST
and a decrease in OLR from the earlier to the latter period (Figure S3).
Besides the sources of the Rossby wavetrain, upper-level zonal wind also
influences the propagating direction of the wavetrains by generating a Rossby
waveguide. Climatological 200-hPa zonal wind display westerly winds south of $20^{\circ}$S
with the largest wind speed around $50^{o}$S (Figure 5a, b). Relative to the 1979-1999
period, stronger westerly winds at southern mid-latitudes during the 2000-2020 period
help the wavetrain go into the waveguide and let it propagate eastwards and
northeastwards into Southern Atlantic Ocean (Figure 4c). Conversely, during the
1979-1999 period, weaker westerly winds do not facilitate the wavetrain excited west
of New Zealand go into the waveguide and the wavetrain only propagate
southeastwards into the Ross and Amundsen Seas (Figure 4a).
In the past century, eastern Pacific ENSO events outnumbered the central Pacific
ENSO events until near the end of the century, when the trend was reversed with the
central Pacific events becoming more frequent (Kug et al., 2009; Freund et al., 2019).
This shift in ENSO regime is reflected in Figure 2. According to Rodrigues et al.
(2015), the two types of ENSO events trigger different atmospheric variability modes
in the Southern Hemisphere represented by the Pacific South American (PSA) pattern.
The central Pacific events trigger the third leading mode (PSA2; the first and second
leading modes being AAO and PSA1, respectively) which, by modulating the
strengths and position of the SASH, connects the tropical Pacific to the Atlantic. The
teleconnection is absent during eastern Pacific ENSO events because these events





trigger PSA1. This helps explain the differences in the SASD-ENSO relationship
between the two periods. The regressions of the SASD index onto the SST anomalies
show that the spatial patterns of the Pacific SST anomalies corresponding to the
positive phase SASD bear a resemblance to the eastern Pacific La Niña state over the
former period, and to a central Pacific La Niña state over the latter period (Figure 6).
The difference of the two patterns (2000-2020 minus 1979-1999) resembles a typical
central Pacific La Niña state. It is, therefore, plausible that the stronger ENSO-SASD
relationship during the recent two decades compared to the proceeding two decades
could be related to the shift in the ENSO regime around the turn of the century from
more frequent eastern Pacific El Niño to more frequent central Pacific El Niño.

To confirm the relationship between the OLR anomalies in the key region and the

SASD anomalies, we conducted an idealized numerical experiment using the CAM5
atmospheric model. In this experiment, we artificially increase the SST by 2 ℃ in the
key region (20 ℃S-40 ℃S, 180 °-140 ℃W), which corresponds to negative OLR anomalies
(as depicted in Figure 2c and 3c). Figure 4S illustrates the anomalous 200-hPa
geopotential height relative to the results from the control experiment, where SST and
sea ice conditions followed the climatological annual cycle. Notably, a wavetrain is
observed from the southern Pacific Ocean to the southern Atlantic Ocean.
Additionally, the experiment shows a weakened South Atlantic Subtropical High
(SASH) over the southern subtropical Atlantic Ocean. The corresponding 1000-hPa
height and wind field display an anomalous low pressure and cyclonic circulation
(Figure 5S), indicating a negative phase of the SASD. Although the OLR anomalies





are not entirely equivalent to SST anomalies, these numerical modeling results
provide further evidence of the negative correlation between the OLR anomalies in
the key region and the SASD.

It is important to note that the differences in the SST anomalies in the equatorial

Pacific (as depicted in Figure 2c), which are indicative of varying ENSO conditions
(i.e., a shifting from eastern to central Pacific El Nino regime around the turn of the
century), laid the background for the OLR anomalies in the key region. Therefore,
these differences played a critical role in modulating the SASD-ENSO teleconnection.

Two additional numerical experiments were conducted, where a +2 ℃ SST

anomaly is introduced in the Niño 4 region over the central tropical Pacific (5°N-5°S,
160°E-150°W) and the Niño 3 region over the eastern tropical Pacific (5°N-5°S,
150°W-90°W), respectively, to ascertain the distinct impacts of eastern and central
Pacific El Niño events on the ENSO-SASD relationship.

As depicted in Figures S6 and S7, in contrast to the idealized experiment

involving SST anomalies over the eastern Pacific Ocean, the idealized experiment
with the SST anomalies over the central Pacific Ocean resulted in geopotential height
anomalies over the southern Atlantic Ocean shifting westward. This westward shift
induced a surface cyclonic circulation, which favored the development of the negative
phase of the SASD mode.

These numerical modeling results provide further confirmation that central Pacific

El Niño events occurring more frequently after 2000 have a more pronounced impact
on the ENSO-SASD relationship.





**4. Conclusion and discussion**
We have revisited the teleconnection of the SST variability in the subtropical
Atlantic Ocean and tropical Pacific Ocean, represented by the SASD and Niño 3.4
indices, respectively. We have showed that SASD and Niño 3.4 are significantly
correlated (r = -0.51, p < 0.01) over the past four decades, with a multi-decadal
variability, which contradicts the weak simultaneous correlations between the two
indices previously suggested in the literature (Venegas et al., 1997; Fauchereau et al.,
2003; Hermes and Reason, 2005; Kayano et al., 2013; Rodrigues et al., 2015). We
have also demonstrated a strengthening of their correlations over the recent two
decades (r = -0.77, p < 0.01) from the preceding two decades (r = -0.32, p > 0.05), and
this significant change in the correlation strength holds true when using either the
Niño 4 index or Niño 3 index in place of the Niño 3.4 index. However, we
acknowledge that the higher frequency of central El Nino events in recent years may
have contributed to the stronger correlations. We have further confirmed the existence
of the multi-decadal variability of the ENSO-SASD relationship through analysis of
century-long SST time series.
Furthermore, we have demonstrated that the changes in the relationship between
SASD and ENSO from the earlier two decades to the recent two decades are not only
associated with the shift in ENSO regime but may also be directly linked to
differences in the anomalous SST and convective activity reflected in OLR anomalies
in the crucial region (20 °S-40 °S, 180 °-140 °W) of the central South Pacific Ocean.
These variations, together with differences in upper-level zonal winds, cause





wavetrains triggered by varying convective activities to propagate along distinct paths,
inducing varying responses in the anomalous MSLP, surface wind, and downward
longwave radiation fields, ultimately resulting in differences in SST anomalies in the
subtropical Atlantic Ocean.
The significant strengthening of the SASD-ENSO relationship in recent decades
may also be related to the sudden phase reversal of the interdecadal-scale modes in
the Pacific and the Atlantic Oceans near the end of the last century (Yu et al., 2017).
In particularly, the Pacific Decadal Oscillation (PDO) and the Interdecadal Pacific
Oscillation (IPO) shifted from positive to negative phase while the opposite occurred
to the Atlantic Multidecadal Oscillation (AMO) around 2000. Salinger et al. (2001)
noted that IPO has a strong influence on the relationship between the ENSO index and
South Pacific precipitation. The positive phase of IPO favors the negative phase of
SASD on interdecadal time scales (Lopez et al., 2016). How IPO (PDO) may
influence the SASD-ENSO relationship remains to be explored in future studies.
Additionally, the tropical passage between the tropical Pacific and Atlantic Oceans
may also influence their relationship (Giannini et al., 2001; Seager et al., 2019). Ham
et al. (2021) found that the SASD mode influences ENSO through tropical Atlantic
SST anomalies and the zonal Walker circulation. Our results further highlight aspects
of the complexity of the interactions between the South Atlantic and Pacific SST
variations. Further exploration of the influence of the IPO and PDO on the
SASD-ENSO relationship, as well as the tropical passage between the Pacific and
Atlantic Oceans, is needed to gain a deeper understanding of the complex interactions





between these coupled atmosphere-ocean systems.
Our study focuses on the simultaneous SASD-ENSO relationship, the SST dipole
structure of the SASD during austral summer (JFM) also may be associated with the
ENSO forcing in previous spring (OND) and winter (JJA) due to the delayed response
of the ocean mixed layer (Saravanan and Chang, 2000; Fernandez and Barreiro,

2022).

In addition to natural variability discussed here, global climate change also may
influence the SASD-ENSO relationship, because global warming may change the
characteristics of ENSO and atmospheric teleconnections related to ENSO
(Martń-Gómez et al., 2020a; Martń-Gómez et al., 2020b; Martń-Gómez and
Barreiro, 2020; Cai et al., 2021;).
Our study contributes to the growing body of knowledge on the interactions
between the South Atlantic and Pacific SST variations, which have a strong influence
on South American and southern African rainfall patterns (Kayano et al., 2013). The
knowledge gained from our study may also improve possibilities for sub-seasonal to
seasonal predictions of precipitation in these regions.

**Code and data availability**
The monthly SST data from the U.S. NOAA Extended Reconstructed Sea Surface
Temperature (ERSST) version 5 (ERSST v5) are available online
(https://www1.ncdc.noaa.gov/pub/data/cmb/ersst/v5/netcdf/). The ERA5 reanalysis
data are available from the below website (https://doi.org/10.24381/cds.6860a573).
The monthly OLR data are derived from the website



(https://psl.noaa.gov/data/gridded/data.uninterp_OLR.html). The monthly SST and
OLR data from 20 models of CMIP6 for SSP1-2.6, SSP2-4.5, and SSP 3-7.0 scenarios
are derived from the website (https://aims2.llnl.gov/search).
Code is available upon request to corresponding author.
**Acknowledgements**
We thank the European Centre for Medium-Range Weather Forecasts (ECMWF) for
the ERA5 data. This study is financially supported by, the National Key R&D
Program of China (2022YFE0106300), and the European Commission H2020 project
Polar Regions in the Earth System (PolarRES; Grant101003590).
**Author contributions**
The research was designed by Lejiang Yu, who also analyzed the data and wrote the
initial draft. Shiyuan Zhong contributed significantly to the writing during both the
initial submission and revision stages. Timo Vihma provided valuable consultation to
the research, while Cuijuan Sui helped with the analysis of sea ice data. Bo Sun
provided comments and played a key role in securing funding for the research. All
authors have reviewed and contributed to the final manuscript
**Competing interests**
The authors declare no competing interests.

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



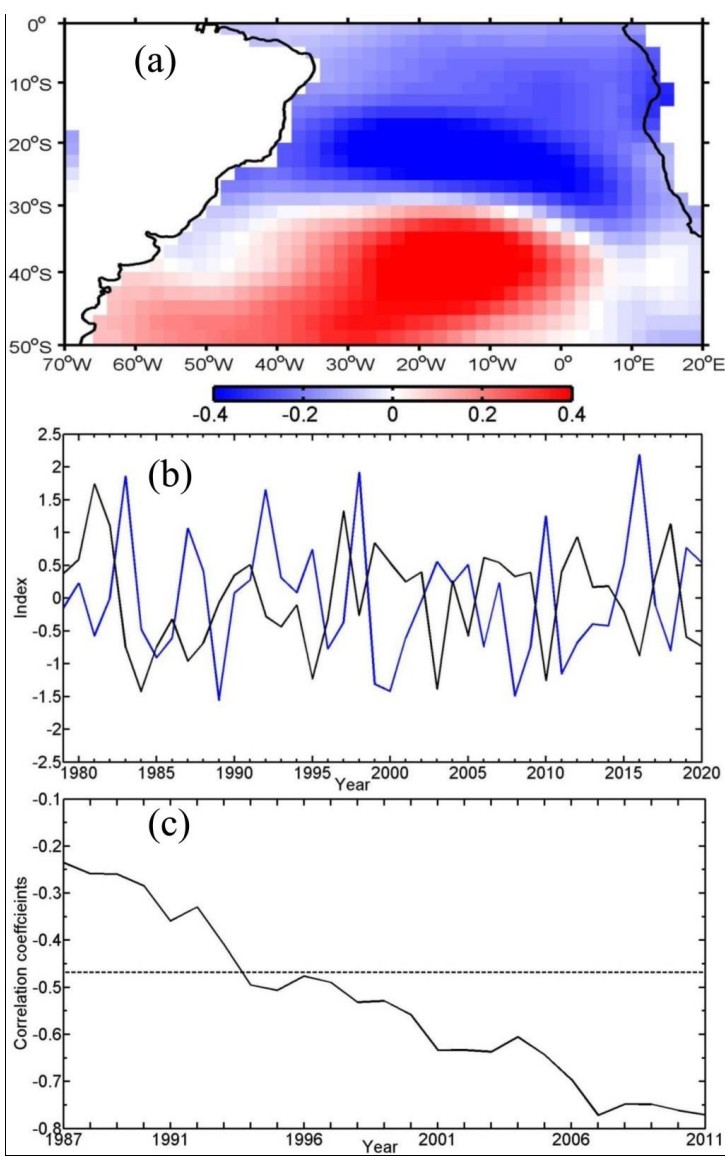

Figure 1. The spatial pattern of SST for the positive SASD phase (a), the time coefficients of the detrended Niño 3.4 (solid blue lines) and SASD (solid black lines) indices in austral summer (JFM) (b), and the 18-year moving correlations between the two indices (black sold line) (c).The dashed line in (c) denotes the correlation coefficients with the above 95% confidence. The number of abscissa denotes the middle of 18-year sliding window. For example, 1987 is the middle year of 1979-1996 period.

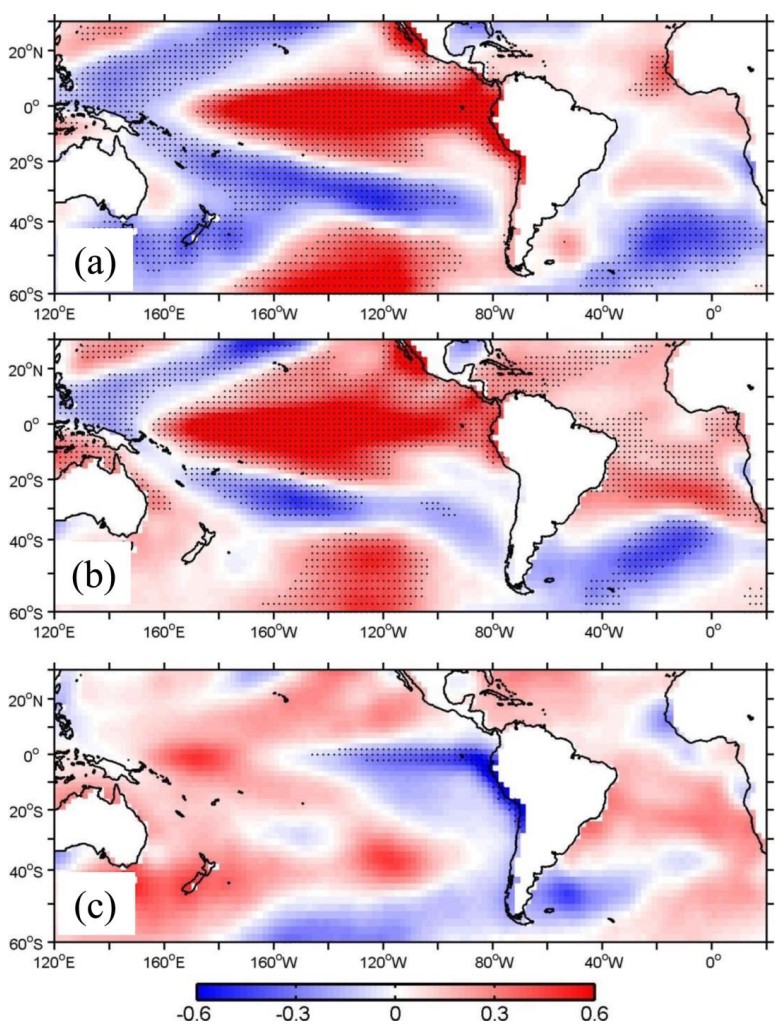

Figure 2. Regression maps of SST anomalies ($^{o}$C) onto the Niño 3.4 index over the 1979-1999 period (a) and the 2000-2020 period (b) and the difference between them (b-a) (c). Dotted regions denote the above 95% confidence level.



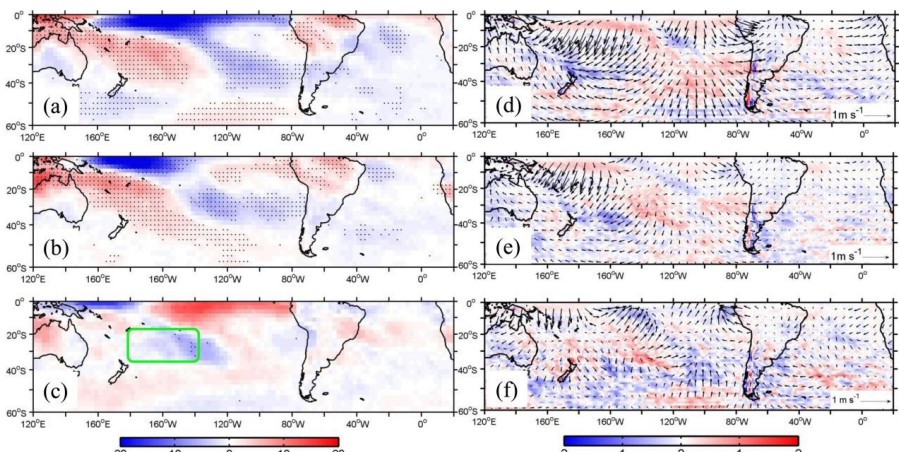

Figure 3. Regression maps of anomalous fields of OLR (W m$^{-2}$) (a), (b), (c), Rossby wave source (RWS) ($10^{-10}$ s$^{-2}$) and 200-hPa divergent wind (vector) (d), (e), (f) onto the Nino 3.4 index over the periods of 1979-1999 (a, d), 2000-2020 (b, e) and the differences between the two periods (latter minus former) (c, f). Dotted regions in panels (a), (b) and (c) denote the above 95% confidence level. The green box in panel c indicates the key region of OLR.



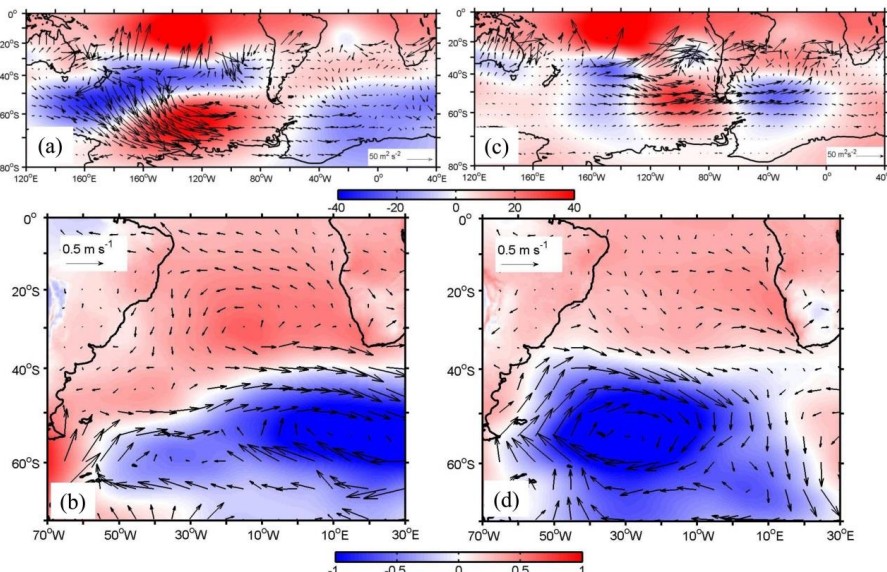

Figure 4. Regression maps of the anomalous fields of WAF (vector) and 200-hPa geopotential height (gpm) (a, c), and mean sea level pressure (hectopascal) and surface wind (vector) (b, d) onto the Nino 3.4 index over the periods of 1979-1999 (a, b) and 2000-2020 (c, d).



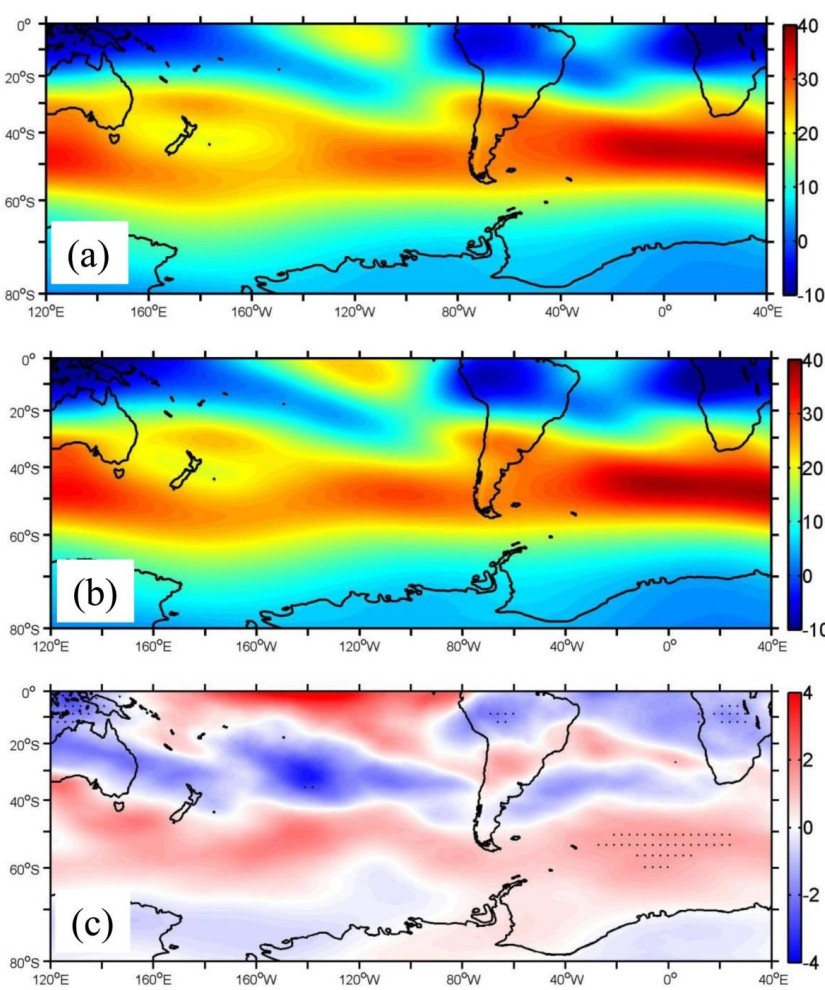

Figure 5. Climatological summertime 200-hPa zonal wind speed (m s$^{-1}$) for the 1979-1999 period (a), for the 2000-2020 period (b) and their difference (c). Dotted regions in panel c denote above 95% confidence level.



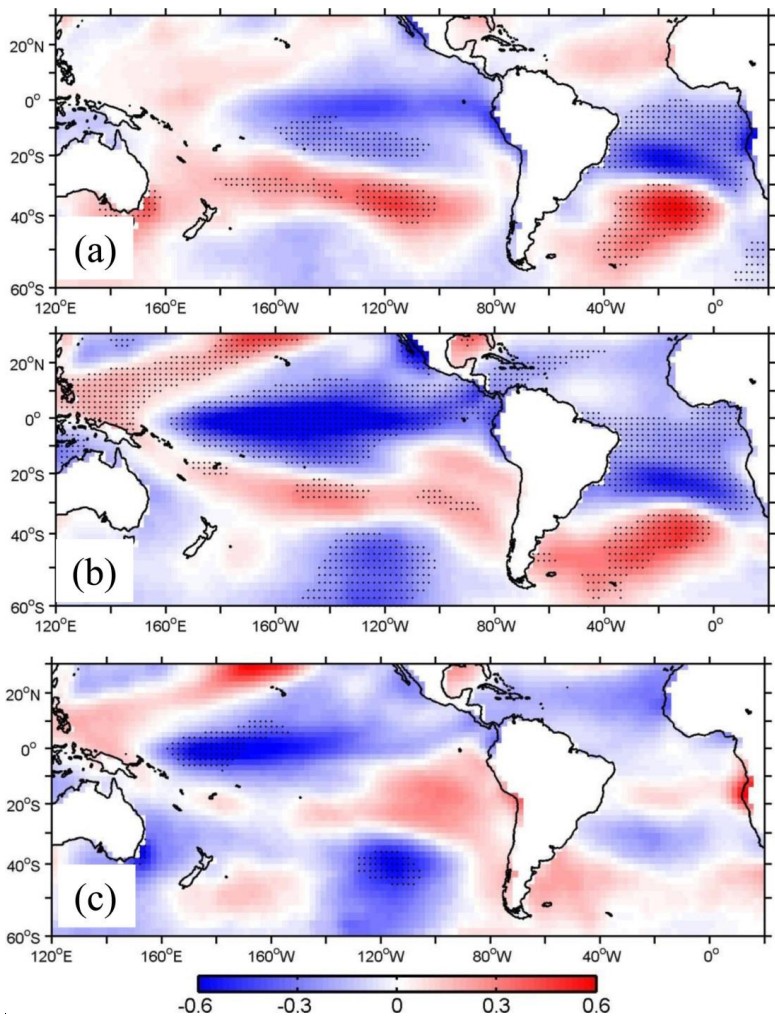

Figure 6. Regression maps of SST anomalies (°C) onto the SASD index for austral summer over the periods of 1979-1999 (a), 2000-2020 (b), and the differences between them (b-a) (c). Dotted regions denote the above 95% confidence level.