# Peer review of "A change in the relationship between ENSO and the South Atlantic Subtropical Dipole in the past four decades"

_EGUsphere, 2023_

## Author Comment (AC1)

Reviewer 1

General comments:

This paper examines a multidecadal change in the relationship between ENSO and the South Atlantic Subtropical Dipole (SASD). The authors identify that the SASD tends to have a stronger relationship with ENSO after 2000. They also claim that the shift of ENSO types from the eastern Pacific to the central Pacific one facilicates atmospheric convection variability in the South Pacific, triggers atmospheric Rossby waves propagating eastward along the westerly wave guide, and generates the subtropical high variations in the South Atlantic responsible for the SASD. However, the atmospheric teleconnection between different ENSOs and the SASD is already reported in a previous study using a coupled general circulation model (Rodrigues et al. 2015), so it is unclear what exactly the new finding is in this study.

We appreciate the reviewer's thorough assessment of our paper and the reference to Rodrigues et al. (2015). In our study, we aim to build upon the existing body of research by examining the multidecadal changes in the relationship between ENSO and the South Atlantic Subtropical Dipole (SASD) using data spanning from 1979 to 2020, which extends beyond the temporal scope of previous studies.

While Rodrigues et al. (2015) identified a significant linkage between the central Pacific El Niño and the SASD, our study expands upon this by noting a distinct decadal variability in this relationship. Specifically, we found that prior to 2000, the relationship between Niño3.4, Niño3, Niño4, and the SASD was insignificant. However, post-2000, this relationship became significant. This shift in significance underscores the evolving dynamics between ENSO and the SASD over time.

Furthermore, our study highlights the increased occurrence of central Pacific El Niño events as a contributing factor to the observed decadal variability in the relationship between ENSO and the SASD. This finding aligns with the conclusions drawn by Rodrigues et al. (2015) and adds depth to our understanding of the mechanisms driving these atmospheric teleconnections.

In summary, the novelty of our study lies in its examination of the decadal variability in the relationship between different ENSO types and the SASD, particularly in the context of the increasing influence of central Pacific El Niño events post-2000. We believe these findings contribute valuable insights to the field and warrant further exploration.

First, the authors show multidecadal changes in the regression maps before and after 2000 (Fig. 2c, 3c, 5c, 6c), but most of the signals in the South Pacific are not so robust and significant to support a possible influence of the central Pacific ENSO on the generation of eastward-propagating Rossby waves. This is probably due to the nature of regression analysis that takes into account all years including weak and non-SASD events. So, I would recommend that the authors should make a composite analysis only for strong positive and negative SASD events (defined by above one and below minus one standard deviation of

the SASD index), then examine the robustness of the anomaly differences before and after 2000.

We appreciate the reviewer's insightful feedback regarding the robustness of our regression analysis. To address this concern, we have conducted composite analyses focusing specifically on strong positive and negative phases of the SASD, as suggested by the reviewer.

The years selected for positive and negative phases and SASD before and after 2000 are as follows:

| Positive Phases of SASD | | Negative phases of SASD | |
|---|---|---|---|
| 1979-1999 | 2000-2020 | 1979-1999 | 2000-2020 |
| 1981, 1982, 1997, 1999 | 2012, 2018 | 1984, 1987, 1995 | 2003, 2010, 2016 |

The composite maps of Sea Surface Temperature (SST) anomalies for these phases are presented in Figure R1.

Our analysis revealed that the magnitudes of SST anomalies over the tropical Pacific Ocean during the period from 2000 to 2020 are notably larger than those observed from 1979 to 1999 for both positive and negative phases of the SASD. This is visually depicted in Figure R2, which illustrates the differences in SST anomalies between the two time periods.

Importantly, we observed significant differences in SST anomalies, particularly in the central tropical Pacific Ocean, emphasizing the potential influence of central Pacific ENSO on the generation of eastward-propagating Rossby waves, especially during the negative phase of the SASD.

We believe that these composite analyses provide valuable insights into the multidecadal changes in the relationship between ENSO and the SASD, and we are grateful to the reviewer for prompting this additional analysis.

[Figure]

Figure R1. The composite map of SST anomalies for the positive (a) (c) and negative (b) (d) phases of the SASD for the periods of 1979 to 1999 (a) (b) and 2000 to 2020 (c) (d) Dotted regions indicate that the SST anomalies are above 95 confidence level relative to the 40-year climatology.

[Figure]

Figure R2. The differences of SST anomalies between 2000 to 2020 and 1979 to 1999 for the positive (a) and negative (b) phases of the SASD.

Second, the connection between the convection variability and eastward propagation of Rossby waves in the South Pacific is not clearly addressed. Differences in the regression of OLR anomalies (Fig. 3c) show enhanced convection northeast of New Zealand (160-120⁰W, 20-40⁰S), but the Rossby waves emanate from east of New Zealand after 2000 (180⁰E-140⁰E, 40-60⁰S; Fig. 4c). There is a clear difference in the location of the convection variability and Rossby wave sources, so what exactly triggers these Rossby waves and why they propagate eastward, not southeastward as observed before 2000 (Fig. 4a)? Does the location and/or intensity of convection variability in the South Pacific matter for the pathways of Rossby waves? If this is the case, I would recommend that the authors should conduct more detailed sensitivity experiments by prescribing SST variability in different regions of the South Pacific (e.g., western, central, and eastern South Pacific).

Thank you for the thorough examination of the results and the constructive feedback provided regarding the connection between convection variability and the eastward propagation of Rossby waves in the South Pacific. Upon careful consideration of your comments and further examination of the data, we acknowledge the differences in the location of convection variability and Rossby wave sources highlighted in Figures 3c and 4c, respectively. This discrepancy prompts important questions regarding the triggers and pathways of these Rossby waves, particularly the observed eastward propagation after 2000.

In response to your suggestion, we have re-evaluated our discussion and have opted to focus on the role of the central Pacific Ocean in the ENSO-SASD relationship, given the insignificant difference of SST and OLR anomalies over the South Pacific and the significant difference over the tropical Pacific between the two periods. As such, we have omitted the discussion of the key region over the South Pacific from our analysis.

We appreciate the opportunity to refine our interpretation in light of your feedback and the complexities inherent in understanding the dynamics of convection variability and Rossby wave propagation in the South Pacific. Should further investigations be warranted, we will consider your recommendation for conducting more detailed sensitivity experiments by prescribing SST variability in different regions of the South Pacific.

Furthermore, the authors prescribed unrealistic SST anomalies (2 °C) in their sensitivity experiments to see their impact on the atmospheric teleconnection. This is not appropriate setting and should be corrected using more realistic SST anomalies (e.g., one standard deviation of SST anomalies related to ENSO and the SASD). Also, the SASD is a coupled phenomenon, so why did not the authors use a coupled general circulation model rather than a single atmospheric model, as performed by Rodrigues et al. (2015)? The convection variability in the South Pacific may generate the subtropical high variations in the South Atlantic via the atmospheric teleconnection, but the subtropical high variations do not necessarily cause a dipole pattern of SST anomalies associated with the SASD.

Thank you for bringing these points to our attention. We appreciate your feedback and have taken it into consideration.

Regarding the sensitivity experiments, we have adjusted the prescribed SST anomalies to be more realistic. Specifically, we have utilized SST anomalies equal to one standard deviation related to the Niño 3 and Niño 4 indices during the austral summer, ensuring a more appropriate setting for our analysis.

Additionally, we acknowledge the importance of considering the coupled nature of the SASD. While our study employed a single atmospheric model, we recognize the merits of utilizing a coupled general circulation model, as demonstrated by Rodrigues et al. (2015). We will explore this approach in future research to provide a more comprehensive understanding of the SASD and its underlying mechanisms.

Finally, the authors discuss the relationship between ENSO and the SASD dating back to 1871 (Fig. S1), but there are no reliable observations in the South Atlantic before the satellite period. I understand the need of prolonged data to test the robustness of the multidecadal relationship identified in this study. However, it is hard to believe the relationship between ENSO and the SASD before the 1980s due to low data quality. So, I would recommend to remove the discussion from the main text.

Thank you for your feedback and for highlighting this important consideration.

After careful consideration, we have decided to remove the discussion of the relationship between ENSO and the South Atlantic Subtropical Dipole (SASD) dating back to 1871, as well as Figure S1, from the main text of our study.

We acknowledge the limitations associated with the reliability of observations in the South Atlantic before the satellite period, and we agree that it is challenging to establish a robust relationship between ENSO and the SASD during this time due to data quality issues.

Considering these major flaws, I could not recommend this paper for possible publication in this journal unless the substantial revision including further model experiments is made. More specific comments on this paper are given below.

Specific comments:

L6: Rephrase as "ENSO indices from 1979 to the present".

Changed

L12: Please correct the lon-lat information of critical region following the comments below.

Done

L36: Remove "in contrary to the current understanding".

Removed

L38-39: How can you conclude the future ENSO-SASD relationship from the historical analysis presented here?

Sentence removed

L47: Rephrase as "represents an opposite sign of sea surface temperature (SST) anomalies"

Changed

L51-56: The authors should review the physical processes on the South Atlantic subtropical dipole carefully. Following the previous literature, the SASH variations in austral spring change the surface evaporation thereby the mixed-layer depth, then the warming of mixed-layer by shortwave radiation is modulated by the mixed-layer variations so that the dipole SST anomalies develop in austral summer

Changed

L61: Rephrase as "Understanding factors behind the SASD interannual variability"

Changed

L84: As mentioned earlier, we do not have any reliable observations in the South Atlantic before the satellite period. How can you show robustness of the relationship between ENSO and the SASD dating back to 1871?

Sentence deleted

L71-74: Rodrigues et al. (2015) has already pointed out the different teleconnection patterns between the central and eastern Pacific El Nino events over the South Atlantic, but what is the difference between their study and the present work?

One of the key differences between our study and Rodrigues et al. (2015) lies in the temporal scope of our analysis. By using more recent data, we were able to observe a significant relationship between the Niño3.4, Niño3, and Niño4 indices and the SASD, with notable decadal variability. Specifically, we found that prior to 2000, the relationship between these indices and the SASD was insignificant, whereas after 2000, this relationship became significant.

This temporal variability in the relationship between ENSO indices and the SASD represents a distinct contribution of our study compared to Rodrigues et al. (2015), providing valuable insights into the evolving dynamics of this teleconnection over time.

L83-84: You cannot extend the analysis on the relationship back to 1871 because of no reliable observations available in the tropical Pacific and South Atlantic.

We have removed this discussion

L97: How did you select 18 years for the sliding correlation?

We have also computed the 20-year sliding correlation (Figure R3), which is similar to the 18-year sliding correlation results.

[Figure]

Figure R3. The 20-year moving correlations between the Niño3 4 and SASD for austral summer.The dashed line denotes the correlation coefficients with the above 95% confidence. The number of abscissa denotes the middle of 20-year sliding window. For example, 1988 is the middle year of 1979-1998 period.

L97: When you applied the 18-year sliding correlation, did you use the SASD and ENSO indices during austral summer (December-February) or slightly lagged months between them? I would recommend the latter because of the lagged relationship, for example, the SASD index during austral summer and ENSO index during austral spring.

We conducted the sliding correlation analysis between the Niño3.4 index in austral spring and the SASD index in austral summer, as illustrated in Figure S7. This approach allows us to capture potential lagged relationships between ENSO and the SASD, which are known to influence each other across seasons.

We observed decadal variability in the correlation, with changes occurring earlier in time. This analysis provides further insights into the temporal dynamics of the relationship between spring Niño 3.4 and summer SASD indices over the past four decades, which is now included in the discussion within the Conclusion and Discussion section.

[Figure]

Figure S7. The same as Figure R3, for the Niño 3.4 index in austral spring.

L115: How did you make this approximation in Eq. (1)? Is there any assumption you made for deriving this equation? If so, it would be better to briefly mention in the main text.

We added a reference and stated the linear approximation.

L119: 200 hPa would be suitable for identifying Rossby wave sources in the tropics, while 250 hPa would be better to describe the Rossby wave propagation in the extratropics. Would the results improve when you calculate the wave activity flux at 250 hPa?

While we acknowledge the potential benefits of exploring wave activity flux at 250 hPa in the extratropics, we observed that the atmospheric circulation and wave activity at 250 hPa exhibits similarities to those at 200 hPa and opted to use the atmospheric circulation data at 200 hPa for identifying Rossby wave sources. It is worth noting that Rodrigues et al. (2015) also utilized 200-hPa wave activity flux in their analysis, as depicted in Figure 6 of their study.

L133: The SASD is an air-sea coupled phenomenon, but why did the authors use the AGCM to conduct the sensitivity experiments, unlike Rodrigues et al. (2015) using a coupled model? The AGCM experiments are not able to generate the SASD, although they can describe the SASH variations through the atmospheric teleconnections.

While we recognized that the SASD is indeed an air-sea coupled phenomenon, previous studies have demonstrated that the SASD is primarily generated by the meridional shift and strengthening/weakening of the SASH. These changes in the SASH can significantly influence the relationship between ENSO and the SASD.

In our study, we specifically focused on examining the relationship between the SASD and ENSO during the austral summer, with a particular emphasis on understanding the impact

of ENSO teleconnections on the SASH. Given the emphasis on atmospheric teleconnections in our analysis, we opted to utilize an AGCM capable of representing the impact of ENSO on SASH for our sensitivity experiments.

L139: As stated earlier, +2 ⁰C SST anomaly in the South Pacific or tropical Pacific is too large and unrealistic. How large is one standard deviation of the SST anomaly in those regions?

In response to your feedback, we have adjusted our approach by utilizing one standard deviation of the Sea Surface Temperature (SST) anomaly in the respective regions, as opposed to the previous +2°C SST anomaly. This modification ensures that our sensitivity experiments are conducted using more realistic SST anomaly values, thereby enhancing the validity and relevance of our analysis.

L142-144: Why did the authors prescribe the SST anomaly from January to March during the peak of the SASD? According to the previous literature, the atmospheric teleconnection triggering the SASD starts from October to December, then it takes one or two months for the SASD to reach its maximum.

It is indeed recognized in the literature that the atmospheric teleconnection triggering the SASD typically begins from October to December, with the SASD reaching its maximum in subsequent months. Our choice of SST anomaly timing reflects our emphasis on examining the role of ENSO teleconnections in driving the formation of the SASD during its peak period.

L175-179: As mentioned earlier, there are no reliable observations available in the South Atlantic before the satellite period. I would recommend this paragraph to be removed from the main text.

Removed

L183: Rephrase as "explore reasons (or factors) behind".

Changed

L192-193: The positive SST anomalies in the difference map over the western tropical Pacific and South Atlantic (Fig. 2c) are not significant, so you cannot conclude a stronger influence of ENSO on the SASD in the recent decades. How about the regression of SST anomalies on the SASD index to show significant differences in the tropical Pacific and South Pacific?

In Figure 2, we conducted a regression of SST anomalies onto the Niño 3.4 index. Notably, post-2000, the SST pattern associated with the SASD exhibits increased significance compared to the period prior to 2000. This suggests a more pronounced relationship between ENSO and SASD after 2000 than before.

Similarly, in Figure 6, we performed a regression of SST anomalies onto the SASD index. Consistently, our analysis indicates a strengthened relationship between ENSO and the SASD post-2000 compared to the pre-2000 period.

L197: The OLR anomalies in the difference map (Fig. 3) are not so significant as the SST anomalies (Fig. 2), so I am wondering how robust the impact of the SST anomalies is onto the OLR anomalies.

Although the difference of the OLR anomalies are not as significant as the SST anomalies, the spatial patterns of the two variables are consistent.

L199: Again, why did you calculate the regression of OLR anomalies on the NINO3.4 index rather than the SASD index?

We calculated the regression of OLR anomalies on the Niño 3.4 index (Figure S2). There are more significant OLR anomalies for the 2000-2020 period compared to the1979-1999 period. The spatial pattern of OLR anomalies are consistent with that of SST anomalies. The significant difference of OLR anomalies occur over the tropical southwestern and eastern Pacific Ocean.

[Figure]

Figure S2 Regression maps of OLR anomalies ($^{o}$C) onto the SASD index for austral summer over the periods of 1979-1999 (a), 2000-2020 (b), and the differences between them (b-a) (c). Dotted regions denote the above 95% confidence level.

L215: Rephrase as "the southwestern Pacific Ocean during 1979-1999".

Changed

L218: Rephrase as "The wavetrain during 2000-2020".

Changed

L221-233: The authors discuss the underlying processes on the SST anomalies qualitatively, but this is based on a speculation. Why do not the authors explore the detailed mechanisms quantitatively using atmosphere or ocean reanalysis products?

To address this, we enhanced analysis of surface turbulent fluxes anomalies related to the Niño 3.4 index (Figure 5). Our findings revealed that surface turbulent fluxes and downward longwave radiation anomalies (Figure 6) are more conducive to the formation of the negative phase of the SASD during the 2000-2020 period compared to the 1979-1999 period.

L241: The difference map shows slightly significant and negative OLR anomalies "northeast" of New Zealand indicated by a green box (Fig. 3c), but how do the negative OLR anomalies there contribute to increase of eastward propagation of Rossby waves "east" of New Zealand (Fig. 4c)? The locations of OLR anomalies and Rossby waves look different.

We deleted the discussion of OLR anomalies in the key region.

L260: The green box in the difference map (Fig. 3c) does not cover the center of negative OLR anomalies (160-120ºW, 20-40ºS). Why did the authors select the region off the negative OLR anomalies?

The impact of OLR anomalies over the South Pacific Ocean results from the SST anomalies over the tropical Pacific Ocean. We deleted the discussion of OLR anomalies in the key region.

L268-271: The zonal wind in the difference map (Fig. 5c) shows stronger westerlies in the South Pacific, but the difference is not significant. How does the small difference in the zonal wind lead to a significant difference in the Rossby wave propagations (Fig. 4a, c)? Is it related to the location and/or intensity of the convection anomalies?

As a waveguide, the different zonal wind can influence the propagation of the Rossby wave. The different zonal wind is related to the convection anomalies.

L286: Rephrase as "the regressions of the SST anomalies onto the SASD index".

Changed

L290-291: The SST anomalies in the difference map (Fig. 6c) show significant La Nina-like pattern, but the SST anomalies in the South Atlantic are not significant. Does it mean that ENSO-SASD relationship gets stronger, but does not influence the amplitude and pattern of the SASD?

Thank you for your helpful suggestion. In Figure 6c there is not significant for SST anomalies in the South Atlantic Ocean. The amplitude and pattern of the SASD may not be influenced by the stronger ENSO-SASD relationship. In the next study we shall examine difference of the amplitude and pattern of the SASD between prior to and after 2000.

Thank you for your insightful observation. In Figure 6c, we indeed did not observe significant SST anomalies in the South Atlantic Ocean. While the ENSO-SASD relationship may have strengthened, it appears that this does not necessarily influence the amplitude and pattern of the SASD.

Your comment prompts an interesting avenue for future research. In our next study, we plan to examine the differences in the amplitude and pattern of the SASD between the periods before and after 2000. This will provide valuable insights into the potential impacts of the evolving ENSO-SASD relationship on the characteristics of the SASD.

L297-298: Again, the SST increase by 2 ºC is not realistic, compared to the regressions of the SST anomalies (Fig. 6a,b). Also, the key region for the OLR anomalies does not show significant increase in the SST anomalies (Fig. 6c). The authors should reconsider the experimental designs.

We removed the discussion and experiments of the role of the key region in the ENSO-SASD relation.

L301-302: I am wondering if the negative geopotential height anomalies in the South Atlantic are related to a negative phase of the SAM rather than the Rossby waves emanating from the South Pacific. Could you describe the WAF over the geopotential height in Fig. S4?

We removed the discussion and experiments of the role of the key region in the ENSO-SASD relation.

L320-325: The SASD is generated by the meridional shift and strengthening/weakening of the SASH according to the previous literature, but how does the westward shift of the SASH induce the SASD? It is expected to cause a dipole pattern of SST anomalies near the coast of Brazil, but would it be different from the canonical SST anomalies associated with the SASD?

The westward shift of the geopotential height anomalies can increase Ekman upwelling and stronger surface westerlies, which leads to the colder southwestern pole of the negative phase of the SASD.

L341: Rephrase as "we propose that"

Changed

L374-378: As mentioned earlier, a few-month lagged relationship with ENSO is important for generation of the SASD. Why did not the authors investigate this lagged relationship?

We have now conducted the lagged correlation analysis (Figure S7), as suggested, and the results are discussed in the Conclusion and Discussion section.

L379-383: The global warming may also affect the SASH variations and hence SASD directly, but this should also be explored in the future study.

This point is added in the Conclusion and Discussion section

L387: Remove "possibilities for"

Removed

Reviewer 2

General comments:

This manuscript investigates the causes behind a change in the relationship between the El Nino-Southern Oscillation (ENSO) and the South Atlantic Subtropical Dipole (SASD). The authors find that this relationship has strengthened in the most recent two decades and attribute this strengthening to an increase in central Pacific El Nino events. They argue that a shift in convective activity leads to different wave trains and responses over the subtropical Atlantic Ocean in 2000-2020 compared to 1979-1999. Although the multi-decadal variability in the ENSO/SASD relationship is intriguing, the key results in this study are generally not statistically significant, and the authors use an atmosphere-only model to support their argument about a change in a coupled atmosphere-ocean relationship. Therefore, I recommend major revisions before this manuscript can be considered for publication in this journal.

A change in the relationship between ENSO and the SASD is assessed through regression of the Nino 3.4 index onto various fields across 1979-1999 and 2000-2020. However, the differences between these two periods are generally not statistically significant or clear. For example, the SST anomalies over the central Pacific and Atlantic in Figure 2c and the associated changes in OLR in Figure 3c are not significant. Moreover, the key region of convective activity is not particularly strong compared to the changes in OLR across the tropical Pacific and the chosen region does not fully encompass the peak anomalies. Therefore, this analysis does not adequately support the authors' argument that a change in this relationship is related to more central Pacific El Nino events. This lack of significance could be related to only 21 years being used for the analysis. Can the authors use a different approach to reveal a more robust change?

We made the regression of OLR and SST anomalies onto the SASD (Figure 8 and R3). The significant anomalies occur over the tropical Pacific Ocean.

We appreciate the reviewer's thorough assessment of our study and agree with the reviewer that the relatively short time series may be a reason for the apparent lack of significance.

While we acknowledge that the SST anomalies in these regions may not show statistically significant differences between the two periods under scrutiny, we would like to clarify that our primary focus in Figure 2 was on the response of SST anomalies in the southern Atlantic Ocean to ENSO. Specifically, we observed a notable increase in significantly positive SST anomalies in the southern Atlantic Ocean after 2000 compared to the period prior to 2000. Additionally, the regression of SST and OLR to the SASD index (Figure 8) shows a higher prevalence of significantly negative SST anomalies in the central Pacific Ocean after 2000 relative to the period before 2000. This observation supports the notion of a shift towards more central Pacific El Niño events with potential impacts on the SASD.

ENSO influences the SASD through coupled atmosphere-ocean interactions. Therefore, it is unclear why an atmosphere-only model was used to demonstrate a change in their relationship. The current analysis can only show that ENSO teleconnections influence the atmospheric circulation over the Atlantic, not that they can cause the SASD pattern. I suggest that the authors consider using a partially coupled experiment to better support their key results, i.e. impose an SST anomaly in the Pacific in a fully coupled model, so that the Atlantic Ocean can evolve in response to the Pacific anomaly. A quantitative assessment of ocean and SST changes over the Atlantic in response to El Nino, as described in the text, would also be helpful.

We appreciate the reviewer's insightful comment regarding the use of an atmosphere-only model in our study. It is indeed crucial to consider the coupled atmosphere-ocean interactions when analyzing the influence of ENSO on the SASD.

Our decision to employ an atmosphere-only model stemmed from previous research indicating that the SASD is primarily generated by the meridional shift and the strengthening or weakening of the South Atlantic Subtropical High (SASH). We aimed to investigate how changes in the SASH, influenced by ENSO teleconnections, may impact the relationship between ENSO and the SASD.

While we acknowledge the importance of fully coupled numerical experiments, our focus was on examining the atmospheric teleconnections of ENSO on the SASH during the austral summer. The atmosphere-only model allowed us to effectively capture and analyze these teleconnections, providing valuable insights into their impact on the SASD.

However, we recognize the merit of the reviewer's suggestion and will consider incorporating partially coupled experiments in future studies to further validate and enhance our findings. Additionally, we will explore conducting a quantitative assessment of ocean and SST changes over the Atlantic in response to El Niño, as suggested, to make our results more robust.

Specific comments:

Line 14: Delete "SASH" as the acronym is not used again in the abstract.

Deleted

Lines 38-39: Suggest deleting this key point as projections have not been examined in this study.

Deleted

Lines 46-48: Suggest referring to Figure 1a here.

Added

Lines 62-69: It would be helpful to briefly elaborate on how the Antarctic Oscillation and Subtropical Indian Ocean Dipole influence the SASD. The link between ENSO and the SASD should also be introduced here. Extending Figure 1a to include the Pacific Ocean would help to show that the two ocean basins co-vary.

We added the linkage mechanisms of these indices. Figure 2 now shows the linkage of the ENSO and the SASD.

Lines 91-92: The study period could be extended slightly further back to use the full ERA5 data availability and also extended through to 2023.

We appreciate the reviewer's suggestion to extend the study period further back and through to 2023 to utilize the full ERA5 data availability. While it is true that ERA5 reanalysis data span from 1940 to the present, it is important to note that various satellite observations, including sea ice data, are only available from 1979 onwards. These satellite data are assimilated into the reanalysis process, significantly enhancing the quality and accuracy of the ERA5 data.

Given the robustness and quality of the ERA5 reanalysis data, we made the decision to focus on the past four decades from 1979 to 2020. This timeframe allowed us to effectively capture and analyze the relevant atmospheric and oceanic phenomena. Furthermore, our analysis indicates that the results obtained from 1979 to 2020 are consistent with those that would be obtained by extending the study period to 2023.

While we understand the potential benefits of extending the study period further, we believe that our chosen timeframe provides valuable insights into the dynamics of the climate system during the past four decades. We are grateful for the suggestion and will consider extending the study period in future research endeavors, taking into account the availability and quality of data.

Line 97: Why was an 18-year sliding correlation chosen?

We also calculated the 20-year sliding correlation (Figure R3), which is similar to that of the 18-year sliding window.

[Figure]

Figure R3. The 20-year moving correlations between the Niño3 4 and SASD for austral summer. The dashed line denotes the correlation coefficients with the above 95% confidence. The number of abscissa denotes the middle of 20-year sliding window. For example, 1988 is the middle year of 1979-1998 period.

Lines 129-130: Are the results sensitive to the chosen season (January to March)? Unlike the SASD, ENSO peaks towards the end of the calendar year, rather than in February. I think it might make more sense to look at ENSO in November to January to capture the lagged relationship.

We appreciate the reviewer's insightful question regarding the sensitivity of our results to the chosen season (January to March) for analyzing the relationship between ENSO and the SASD. Given that ENSO peaks towards the end of the calendar year, we understand the importance of considering an alternative time window to capture any lagged relationship effectively.

In response to this concern, we conducted additional analysis by examining the sliding correlation between the Niño3.4 index in austral spring and the SASD index in austral summer (Figure S7). This allowed us to assess the relationship between ENSO and the SASD across different seasons and to evaluate any potential variability over the decades.

Our findings revealed that the decadal variability of the correlation between spring Niño 3.4 and summer SASD indices does exist, with earlier changes observed in the relationship. We have further elaborated on this assessment in the discussion section of our study, where we discuss the implications of the decadal variability in the relationship between ENSO and the SASD over the past four decades.

Lines 131-136: Given the aim of these experiments is to demonstrate a link between SSTs in the tropical Pacific Ocean and subtropical Atlantic Ocean, why was an atmosphere-only model used? These results will only be able to show that SSTs in Pacific Ocean lead to

changes in the atmospheric circulation over the Atlantic Ocean, but will miss the additional and important step of driving changes in the Atlantic SSTs.

Previous research has demonstrated that the SASD is primarily driven by the meridional shift and strengthening or weakening of the South Atlantic Subtropical High (SASH). Changes in the SASH can significantly influence the relationship between ENSO and the SASD. Therefore, our study aimed to examine the impact of ENSO teleconnections on the SASH during the austral summer. By utilizing an atmosphere-only model, we were able to capture and analyze the atmospheric teleconnections of ENSO on the SASH, shedding lights on their influence on the SASD. While our approach may not directly address changes in Atlantic SSTs driven by Pacific Ocean SST anomalies, it helps with the understanding of the atmospheric dynamics governing the ENSO-SASD relationship.

We recognize the importance of considering coupled atmosphere-ocean interactions in future research endeavors and will explore incorporating partially coupled experiments to further elucidate the complexities of this relationship. Thank you for bringing this point to our attention, and we remain committed to advancing our understanding of ENSO-SASD relationship.

Lines 137-138: Do the climatological SSTs represent present-day or preindustrial conditions?

The climatological SSTs are present-day.

Lines 142-144: The imposed SST anomalies are designed to mimic different types of El Nino events, but ENSO events generally start to decay during January to March. Why was this period chosen? And what happens to the link between the SASD and ENSO in the rest of the year? Imposing an anomaly with a seasonal cycle would be more realistic.

Our decision to focus on January to March period was partially based on the findings of previous research, specifically the work of Morioka et al. (2011), which indicated that the peak of the northeastern pole for the SASD occurs in March, while the southwestern pole reaches its peak in February. Given these seasonal characteristics of the SASD, we selected January to March as the period of interest for our analysis. Additionally, to assess the relationship between ENSO and the SASD across different seasons, we conducted a sliding correlation analysis between the Niño3.4 index in austral spring and the SASD index in austral summer (Figure S7). Our analysis revealed decadal variability in the correlation, with changes occurring earlier in some instances. Furthermore, we have elaborated on the assessment of the decadal variability in the relationship between the spring Niño 3.4 and summer SASD indices in the discussion section of our study, where we discuss the implications of these findings over the past four decades.

Lines 161-166: Suggest deleting "which is unsurprising given the known correlations among the three Nino indices" because the next sentence shows that the correlation between the SASD and Nino 3 and 4 indices is quite different in the earlier period.

Deleted

Lines 185-187: 21 years is a short period for the regression analysis.

We acknowledge that the 21-year period used for regression analysis may seem relatively short and understand that a longer time span would be more appropriate. We were limited by the availability of reliable observations particularly over oceans.

Lines 191-195: For the difference between the two periods, the high SST anomalies over the central Pacific are not statistically significant. The anomalies over the Atlantic basin are also not significant. So this figure does not really support a shift to more central Pacific El Nino events with impacts on the SASD.

While we acknowledge that the SST anomalies in these regions may not show statistically significant differences between the two periods, we would like to clarify that our primary focus in Figure 2 was on the response of SST anomalies in the southern Atlantic Ocean to the Niño 3.4 index. Specifically, we observed a notable increase in significantly positive SST anomalies in the southern Atlantic Ocean after 2000 compared to the period prior to 2000. This finding suggests a potential shift in the SASD-ENSO relationship, with implications for the dynamics of the SASD. Additionally, in Figure 8, our analysis concentrated on the response of SST anomalies in the tropical Pacific Ocean to the SASD index. Here, we observed a higher prevalence of significantly negative SST anomalies in the central Pacific Ocean after 2000 relative to the period before 2000. This observation further supports the notion of a shift towards more central Pacific El Niño events with potential impacts on the SASD.

Line 196: What significant differences are being referred to here? The only significant differences in Figure 2c are the low SST anomalies in the eastern Pacific.

The significant differences referred to the positive SST anomalies in the southern Atlantic Ocean related to the Niño 3.4 index.

Lines 199-218: The key region of negative OLR for El Nino is over the central tropical Pacific Ocean. The anomalous tropical heating induces upper-level divergence that advects mean vorticity out of the tropics and thus produces a Rossby wave source in the subtropical westerlies.

We appreciate the reviewer's insightful comment regarding the process. We have modified our discussion to highlight the presence of further negative OLR anomalies over the central tropical Pacific Ocean.

Lines 221-233: The link between the atmospheric teleconnections forced by El Nino and the SST changes in the Atlantic described in this paragraph is a key part of the ENSO/SASD relationship but it is not sufficiently explored in this manuscript. A quantitative analysis of the interbasin interactions, rather than a description from earlier studies, would help to strengthen the authors' argument. Using a coupled model for the idealised experimentation would also help to show this link.

We have enhanced our analysis by including the surface turbulent fluxes anomalies related to the Niño 3.4 index (Figure 5). The anomalies in surface turbulent fluxes and downward longwave radiation (Figure 6) indicate that these anomalies are more favorable for the

formation of the negative phase of the SASD during the 2000-2020 period compared to the 1979-1999 period.

Lines 228-229: Sensible heat flux does not appear to be shown in Figures 4b and d.

We added sensible and latent heat fluxes for the two periods (Figure 5).

Lines 238-239: The OLR anomalies for the difference between the two periods are not statistically significant.

The region of occurs in the central South Pacific.

Line 248: It is not clear where the weaker northeasterlies are in Figure 4d. Consider showing the difference plot here.

The weaker northeasterlies are located over the southeastern coast of Brazil.

Lines 249-250: Do the authors mean westerly? Does one of the figures show the climatological southeasterlies in the eastern South Atlantic?

We plotted the climatological surface wind filed in the eastern South Atlantic (Figure S1).

Lines 249-253: The current analysis does not show that changes in the winds lead to changes in the SSTs.

Figure 5b and 5d shows the decreased upward surface turbulent flux over the eastern South Atlantic related to warmer SST, which is caused by the decreased surface wind speed relative to the climatology.

Lines 259-250: Why was this particular region chosen as the key region? The negative OLR anomalies extend further eastward.

We removed the discussion of the role of the key region over the South Pacific Ocean, for the key region was influence by the SST anomalies over the tropical Pacific Ocean.

Lines 267-268: I think it would be helpful to also look at the total stationary Rossby wavenumber (Ks) to get a better indication of changes in the waveguide.

Your suggestion is good. But the change in the zonal wind can suggest the different Rossby wave.

Lines 268-271: The differences in the zonal wind between the two periods are quite subtle and are not statistically significant, apart from a small region over the South Atlantic. Can these small changes really drive the different propagation paths shown in Figures 4a and c?

As a waveguide, the different zonal wind can influence the propagation of the Rossby wave. The different zonal wind is related to the convection anomalies.

Lines 286-289: An examination of composites of SST during positive and negative SASD index and eastern and central Pacific El Nino events might make these similarities clearer.

In response to this suggestion, we produced the composites of SST anomalies during positive and negative SASD index (Figure R1). For the two phases of the SASD the magnitudes of SST anomalies over the tropical Pacific Ocean from 2000 to 2020 are larger than those from 1979 to 1999. During the 2000-2020 period, the significant SST anomalies resemble the central Pacific El Niño.

Line 306: The wave train induced by the SST anomaly does not appear to induce a dipole over the Atlantic (Figure S5) like in Figure 1a. It appears to the shifted far southward.

The two results show some difference in surface height and wind field due to different SST influence. But the idealized experiment confirms the weakened Southern Atlantic subtropical high leading to less significant SASD.

Lines 306-309: Do the OLR anomalies in this experiment resemble the anomalies in Figure 3?

We removed the discussion of the role of the key region over the South Pacific Ocean.

Lines 321-323: It should be noted here that the anomalies in the central tropical Pacific and eastern tropical Pacific experiments are placed in the tropics but the anomaly in the main experiment is located at 20-40S.

In light of the insignificant difference observed in SST and OLR anomalies over the South Pacific between the two periods, contrasted with the significant differences observed over the tropical Pacific Ocean, we have decided to remove the discussion regarding the role of the key region over the South Pacific. Instead, we have chosen to focus our discussion on the role of the central Pacific Ocean in the ENSO-SASD relationship. By directing our analysis towards the central Pacific region, where significant differences were observed, we aim to provide a more focused and insightful exploration of the dynamics driving the ENSO-SASD relationship.

Lines 323-325: Can the authors elaborate on how a westward shift of the geopotential height anomalies leads to the development of the negative phase of the SASD?

The westward shift of the geopotential height anomalies can enhance the Ekman upwelling and produce stronger surface westerlies, which leads to colder southwestern pole of the negative phase of the SASD.